

**Throughfall spatial patterns translate into spatial patterns of soil moisture dynamics – empirical evidence**
Christine Fischer[1,2], Johanna Clara Metzger[1,3], Gökben Demir[1], Thomas Wutzler[4], Anke Hildebrandt[1,5*]
[1] Institute of Geosciences, Friedrich-Schiller-University Jena, Burgweg 11, D-07749 Jena, Germany
[2] Office for Green Spaces and Waters, City of Leipzig, Prager Straße 118-136, D-04217 Leipzig, Germany
3 Institute of Soil Science, University of Hamburg, Allende-Platz 2, 20146 Hamburg, Germany
[4] Max-Planck-Institute for Biogeochemistry, Jena, Germany
[5] Department Computational Hydrosystems, Helmholtz Centre for Environmental Research - UFZ, Leipzig, Germany
* corresponding author
**Abstract**
Throughfall heterogeneity induced by the redistribution of precipitation in vegetation canopies has repeatedly been
hypothesized to affect the variation of soil water content and runoff behavior, especially in forests. However, we
are not aware of any observational study relating the spatial variation of soil water content directly to net
precipitation to confirm modelling hypotheses. Here, we investigate whether throughfall patterns affect the spatial
heterogeneity of soil water response in the main rooting zone. We assessed rainfall, throughfall and soil water
contents (two depths: 7.5 cm and 27.5 cm) on a 1-ha temperate mixed beech forest plot in Germany 2015 - 2016
during the growing seasons in independent high-resolution stratified random designs. Because throughfall and soil
water content cannot be measured at the same location, we used kriging to derive the throughfall values at the
locations where soil water content was measured. We first explore the spatial variation and temporal stability of
throughfall and soil water patterns, and next evaluate the effects of input (throughfall), soil properties (field
capacity and air capacity), and vegetation parameters (canopy cover and distance to the next tree) on soil water
content and dynamics.
Throughfall spatial patterns were related to canopy density. Although spatial auto-correlation decreased with
increasing event sizes, temporally stable throughfall patterns emerged, leading to reoccurring high and lower
input locations across precipitation events. A linear mixed effect model analysis showed, that soil water content
patterns were only poorly linked to throughfall spatial patterns, and it was rather shaped by unidentified but time
constant factors.
Instead of soil water content itself, the patterns of its increase after rainfall corresponded more closely to
throughfall patterns, in that more water was stored in the soil where throughfall was elevated. Furthermore, soil
moisture patterns themselves enhanced or decreased water storage in the soil, and probably fast drainage and
runoff components. Locations with low topsoil water content tended to store less of the input water, indicating
preferential flow. In contrast in subsoil, locations with high water content stored less water. Also, distance to the
next tree and air capacity modified how much water was retained in soil storage.
In this comprehensive study we show that throughfall patterns imprint less on soil water contents and more on soil
water dynamics shortly after rainfall events, therefore only partly confirming previous modelling with data. Our
findings highlight at the same time systematic patterns of times and locations where the capacity to store water is
reduced and water probably conducted quickly to greater depth. Our results indicate that not soil moisture patterns
but rather percolation may depend on small scale spatial heterogeneity of canopy input patterns.

Keywords: throughfall, mixed beech forest, soil water content increase, temporal variation, spatial variation,
pattern



## 1. Introduction

Over the past decades, there has been a raised interest on how water input at the soil surface is affected by vegetation canopies to understand and predict hydrological processes related to vegetation structure and land use change (Western et al., 2004; Savenije, 2004; Murray, 2014; Guswa et al., 2020; Oda et al., 2021). Due to interception losses, the water arriving below the canopy is a smaller amount compared to above (Horton, 1919 and references therein; Carlyle-Moses and Gash, 2011) with implications for the soil water balance (Durocher, 1990; Bouten et al., 1992; Schume et al., 2003; Klos et al., 2014; Metzger et al., 2017) and overall water budget at the catchment scale (Brown et al., 2005; Oda et al., 2021).

Next to interception loss, the contact of precipitation with the vegetation canopy causes spatial redistribution of the incoming water. This leads to characteristic spatial heterogeneity of the dripping (thoughfall) and flowing (stemflow) below canopy precipitation, locally causing enhanced water input to the soil surface. For example, hotspots by dripping points (enhanced water flow from peculiarities in the canopy, Falkengren-Grerup, 1989; Keim et al., 2005; Staelens et al., 2006; Voss et al., 2016) and stemflow hotspots (Levia and Germer, 2015; Carlyle-Moses et al., 2018) are well-documented. The available research suggests that both throughfall patterns and stemflow spatial distributions are reoccurring (Keim et al., 2005; Staelens et al., 2006; Zimmermann et al., 2008; Wullaert et al., 2009; Guswa and Spence, 2012; Metzger et al., 2017; Van Stan et al., 2020).

The observed persistence of spatial patterns of below canopy precipitation has created a strong expectation that those affect patterns of soil water content (Schume et al., 2003; Wullaert et al., 2009; Rosenbaum et al., 2012; Zehe et al., 2010) and hotspots of percolation or preferential flow (Bouten et al., 1992; Schume et al., 2003; Blume et al., 2009; Bachmair et al., 2012) in forests soils. Yet, this is only partly confirmed with observations: For stemflow affected locations, soil moisture microsites have repeatedly been demonstrated (Pressland, 1976; Durocher, 1990; Liang et al., 2007; Germer, 2013; Metzger et al., 2021). Stemflow can create substantial funneling of water to the forest floor and water availability on the forest floor can be locally enhanced 10 to 100 times (Levia and Germer, 2015; Carlyle-Moses et al., 2018; Metzger et al., 2021).

While for stemflow the belowground consequences of input hotspots have been repeatedly confirmed, much less research is available about the role of the less pronounced, but still spatially persistent pattern of throughfall for soil water dynamics. Modelling suggested that throughfall patterns influence the root zone soil moisture pattern (Coenders-Gerrits et al., 2013; Guswa, 2012). However, soil moisture patterns are also influenced by several other factors creating substantial heterogeneity such as heterogeneity of soil properties, local micro-topography, litter thickness or root water uptake (Bouten et al., 1992; Schume et al., 2003; Schwärzel et al., 2009; Gerrits and Savenije, 2011; Rosenbaum et al., 2012; Liang et al., 2017; Molina et al., 2019), and those are typically not fully captured in virtual experiments. In contrast, observation studies found that throughfall and root zone soil moisture were not (Shachnovich et al., 2008; Rodrigues et al., 2022) or only occasionally (Metzger et al., 2017) or weakly (Molina et al., 2019) related. On the other hand, Klos et al. (2014) found a relation below the rooting zone by strategically sampling at high and low throughfall positions, and several authors found indirect evidence by interpreting the change of spatial variation in soil water content (Zehe et al., 2010; Rosenbaum et al., 2012; Metzger et al., 2017) after precipitation events.




In light of the substantial heterogeneity of other influencing factors, one of the reasons for the limited direct
observational evidence of the effect of throughfall on soil water content maybe the lack of studies investigating
the relation between below canopy precipitation and soil water patterns in a dedicated and coordinated fashion.
The characterization of spatial patterns, such as those of throughfall, requires a large number of samplers
(Kimmins, 1973; Lloyd and Marques, 1988; Zimmermann et al., 2010; Van Stan et al., 2020), and the same is
true for below ground observations. Furthermore, a fundamental challenge is that soil water input and soil water
content cannot be assessed at the same location, since the throughfall measurements disturb the infiltration into
the soil. The objective of this study is therefore to compare the patterns of soil water content, soil properties and
throughfall using a dedicated spatially highly resolved sampling design to reveal whether input, next tree
distance or soil properties affect spatial variation in soil water content and soil water response. We used
independent designs for above and below ground observations and applied kriging to derive the throughfall
values at the locations where soil water content was measured. The aims of the study were to a) to explore
spatial heterogeneity and temporal stability of throughfall and soil water content and b) evaluate the influence of
soil properties (field capacity and macroporosity), vegetation parameters (canopy cover, next tree distance) and
input variation (throughfall) on the variation of soil water content and soil water content increase after
precipitation.

## 102  2. Methods

### 104  2.1 Study area

The study was carried out in the Hainich Critical Zone Exploratory (CZE Hainich, see Küsel et al. 2016), run by
the Collaborative Research Centre "AquaDiva". The site is located in Central Germany, in the Hainich National
Park in an unmanaged beech dominated forest. Mean annual temperature are around 7.5 to 9.5 °C, depending on
the position of the small mountain, and the and total annual precipitation drops from 900 to less than 600 mm from
ridge to valley (Küsel et al., 2016). The monitoring site as well as measurements of precipitation and soil moisture
has been described in Metzger et al. (2017), the important parts are repeated here for completeness. The site covers
an area of 1 ha and is situated at 365 m a.s.l.. The study area contains of 581 tree individuals (diameter breast
height ≥ 5 cm), representing a heterogeneous age structure. The soils in this area are dominly luvisols (Schrumpf
et al., 2014; Kohlhepp et al., 2017). The species assemblages consists of 70% European beech trees (*Fagus*
*sylvatica*), as well as Sycamore maple (*Acer pseudoplatanus*), European ash (*Fraxinus excelsior*), European
hornbeam (*Carpinus betulus*), Large-leafed linden (*Tilia platyphyllos*), Norway maple (*Acer platanoides*) and
Scots elm (*Ulmus glabra*). The weathered bedrock is at 15 to 87 cm depth (median depth 37 cm). More details on
the research site are given in Metzger et al. (2017).

### 119  2.2 Precipitation measurements and processing

The precipitation sampling follows the same procedure as given in Metzger et al. (2017). Gross precipitation ($P_g$)
and throughfall ($P_{TF}$) were measured manually using gauges on a per-event basis in spring 2014, 2015, 2016. The
current analysis covers the period from June 18 to July 28 2015 and May 31 to July 14 2016. The installed
throughfall collectors consist of circular funnels (diameter = 12 cm), the opening of which is placed about 37 cm
above the ground surface. A table tennis ball is placed in the opening of the funnel to minimized evaporation.





Throughfall collectors were arranged in a stratified sampling design (Zimmermann et al., 2016). For this, the 1 ha
plot was divided into 100 subplots each 10 m x 10 m (Figure 1) and equipped with two randomly located
throughfall samplers. Of those, we selected 50 point randomly and added another sampler in direct vicinity (0.1 m
distance) creating a "short transect". Furthermore, to 25 randomly selected short transects we added four more
samplers at 0.5, 1, 2, and 3 m from the first to form "long transects". The direction of all transects was also
randomly chosen. In total we sampled n = 350 throughfall positions.
Sampling started 2 h after the end of rainfall by collecting the volume of all sampling containers using graduated
cylinders. Gross precipitation was measured at an adjacent (distance 250 m) open grassland using five funnels of
the same type as the throughfall collectors.

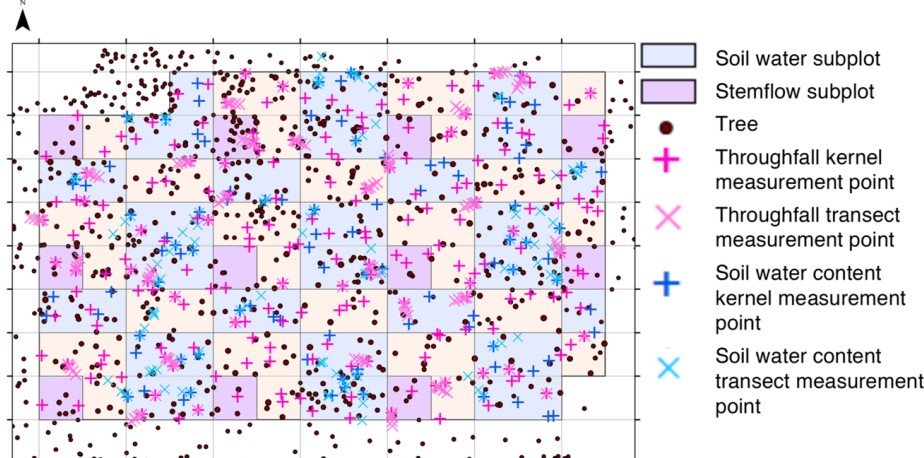


Fig. 1: Experimental set-up in the 1-ha forest plot subdivided by a 10 m x 10 m grid yielding 100 subplots.
Positions of the throughfall samplers (pink crosses) and 49 soil water content subplots (blue) measured in a
stratified random design with transects (see material and methods for more details, Figure from Metzger et al.,

138    2017).


To allow comparison of spatial pattern between events, we calculated a normalized spatial deviation of each
measurement ($\delta P_{\text{TF,i}}$) similarly to Vachaud et al. (1985). Since throughfall is not always normally distributed in
space, we used the median ($\hat{P}_{TF}$) instead of the arithmetic mean for normalization, as already done by Zimmermann
et al. (2008) and Wullert et al. (2009) as follows

$$\delta P_{\text{TF,i}} = \frac{P_{\text{TF,i}} - \hat{P}_{TF}}{\hat{P}_{TF}} \tag{1}$$

where $\delta P_{\text{TF,i}}$ represents the normalized value of the spatially distributed measurements of throughfall ($P_{\text{TF,i}}$) at
locations i for a specific event, and $\hat{P}_{TF}$ the spatial median for that event. To investigate the temporal persistence
of the spatial pattern of throughfall we derived temporal stability plots (Zimmermann et al., 2008; Wullaert et al.,
2009) by ranking the normalized throughfall from minimum to maximum. Additionally, we calculated Spearman
rank correlation coefficients between observations of different events, where high correlations indicate strong



persistence (or temporal stability) of the throughfall pattern. We paired all events falling into a given rain event
class according the Metzger et al. (2017): small: ($P_g \leq 3$ mm); medium (3 mm $< P_g \leq 10$ mm), large ($P_g > 10$ mm).
To relate the general precipitation and soil moisture conditions during the observation period to the average
climate, we compared them with precipitation data from a nearby gauge (Mühlhausen- Windeberg, 20 km to the
northeast) of the German Weather Service (DWD climate data centre, www.dwd.de/cdc, ID 5593).

**2.3 Soil water content measurements**
The soil water measurements were first described in Metzger et al. (2017). Volumetric soil water content was
monitored using a wireless sensor network (SoilNet, Bogena et al. (2010)) equipped with SMT100 frequency
domain sensors (Truebner GmbH, Neustadt, Germany). Overall 210 soil water content measurement points were
distributed in a stratified random design in the blue subplots shown in Figure 1: Within each blue subplot, two
sampling points were placed randomly. Additionally, to a subset of 24 randomly selected points, transects were
added with three additional measurement points (at 0.1, 2.0, and 6.0 m from the position). Furthermore, 40
locations were added as transects near tree stems. At each soil moisture measurement location, sensors were
installed in two depth, e.g topsoil 7.5 cm and subsoil 27.5 cm depth. For this analysis we used the data collected
during the throughfall measurement campaigns from June 18 to July 28 2015 and May 31 to July 14 2016. At each
locations, we used soil moisture measurements an hour preceding the observed rain event ($\theta_{pre,i}$) to characterize
soil moisture and its pattern in the drained state and the maximum soil water content induced by the rain event
($\theta_{post,i}$) to characterize the post event state. We also assessed the soil water content response by calculating the
change of soil water content ($\Delta\theta_i$) for each event and each location with

$$\Delta\theta_i = \theta_{post,i} - \theta_{pre,i} \tag{2}$$

where positive values of $\Delta\theta_i$ indicate water content increase.
Equivalently to throughfall, we calculated the median soil water contents ($\hat{\theta}_{pre}$, $\hat{\theta}_{post}$) as well the relative
deviations ($\delta\theta_{pre,i}$, $\delta\theta_{post,i}$), indicating the spatial pattern of soil water content according to Equation 1. Using the
normalized values of soil water content and throughfall next to the medians in the statistical models (see below)
allowed us to differentiate between spatial patterns and temporal variation across events.

**2.4 Canopy and soil property measurements**
At the time of soil sensor installation, undisturbed soil samples were collected using metal ring cylinders with a
volume of 100 cm³. The distance between the sensor position and the soil sample collection was approximately
0.5 m. Soil properties were treated as if they were measured directly at the soil sensor location i. In order to
determine field capacity ($\theta_{FC,i}$), the samples were first saturated and next let drain in a sand box with a hanging
water column imposing a pressure of −60 hPa for 72 hours and weighed. The soil cores were subsequently dried
for 24 h at 105° C and weighed again to obtain the dry weight $m_{dry,i}$. The volumetric water content at field
capacity ($\theta_{FC,i}$) was derived from the weight difference of the sample at -60 hPa and the dried one, while
assuming a density of water of $D_w = 1$ g cm⁻³. Bulk density ($D_{bd,i}$) was calculated from soil dry weight and
volume. Soil apparent porosity ($\varphi_i$) was calculated from the bulk density and assuming a constant density of the
soil mineral component ($D_m = 2.66$ g cm⁻³)



$$\varphi_i = 1 - \frac{D_{bd,i}}{D_m} \qquad (3)$$

Air capacity ($\theta_{AC,i}$, also called air-filled porosity) was then determined as

$$\theta_{AC,i} = 1 - \theta_{FC,i} \qquad (4)$$

To characterize the canopy density, we counted the number of branches (canopy cover) above the throughfall
samplers in 2014. This data was however not available for soil water measurement locations.

**2.5  Statistical Analysis**
All statistical analysis were processed with R 3.2.3 (Core Team 2016). For the geostatistical analysis (detailed
below) we used the the packages *geoR* (Ribeiro Jr and Diggle, 2001), *georob* (Papritz and Schwierz, 2020) and
*gstat* (Pebesma, 2004; Gräler et al., 2016). Linear mixed effects models were implemented using the package *lme4*
(Bates et al., 2015) and *lmerTest* (Kuznetsova et al., 2017). The variance explained by fixed and random factors
(conditional R²) and by only fixed effects (marginal R², Nakagawa and Schielzeth (2013)) for the final model were
calculated with the *MuMIn* package (Barton, 2020).

**2.5.1    Geostatistical estimation of throughfall**
Throughfall was estimated at the soil water content measurement locations by kriging. The overall procedure for
obtaining the variograms closely follows Zimmermann et al. (2016) with some adaptations taken from Voss et al.
(2016). Important steps and decisions of the exploratory data and geostatistical analysis are shown in Figure S1.

*1. Exploratory Analysis-Test for trends and underlying asymmetry.* First, we determined the skewness using the
octile skew. The octile skew of none of the throughfall events was larger than 0.2 or smaller -0.2 and we therefore
did not transform the data. If a spatial trend existed (p ≤ 0.150), we used the residuals of the spatial regression
model for the coordinates x and/or y instead of the real data in the following.

*2. Variogram estimation by the method-of-moments (MoM).* We calculated the empirical throughfall variogram
using both non-robust and robust estimators (Matheron, 1962; Cressie and Hawkins, 1980; Dowd, 1984; Genton,
1998) using the *sample.variogram* function in the package *georob* in R. For throughfall we chose lags centered at
0.125, 0.375 and 0.75, followed by a step size of 1 m up to 50 m). Next, we obtained a provisional variogram,
which serves for spatial outlier detection in step 3. For this, we fitted three models to the experimental variogram
(spherical, exponential and pure nugget) using *fit.variogram.model* function in the package *gstat* and chose the
model with the lowest Residual Sum of Squares. Then we assessed the fitted model by leave-one-out cross
validation. Based on this we calculated the normalized kriging error ($\Theta_i$,) (Lark, 2000) and compared the
variograms from all mentioned estimators using the estimator with a median of $\Theta$ nearest to 0.455 (Zimmermann
et al., 2008).

*3. Identification and spatial outlier removal.* Before final variogram estimation using residual maximum likelihood
(REML) in step 4, outliers were removed based on kringing and cross validation using the provisional variagram
obtained in step 2. For identifying a spatial outlier at location *i* we used the standardized error of cross validation
$\varepsilon_{s,i}$ (Bárdossy and Kundzewicz, 1990, Lark, 2002). To classify an outlier we used the *Z*-statistics. Sampled points
with $\varepsilon_{s,i} < -2.576$ ($\alpha/2 = 0.005$) were removed (Zimmermann et al., 2016).




*4. Variogram estimation by residual maximum likelihood (REML).* After outlier removal, we applied REML to fit
the theoretical model including spatial trend if necessary, using the *likfit* function in the package *geoR*. We used
the initial estimates from the provisional variogram (step 2) for the parameters sill, nugget and range. The range
relates to the distance over which the observations are still spatially correlated. In the following, we will use the
term correlation length to refer to the effective range, e.g. the distance at which the variogram approaches the sill
to 95%. For example, a high effective range indicates a high spatial correlation between the throughfall collectors.
We checked the reliability of the final model with the statistic $\Theta_i$ (see above).

*5. Kriging.* Using the final variogram from step 4, we applied ordinary kriging to predict throughfall values at the
soil water content measurement locations. Locations where the kriging variance exceeded 95% of the spatial
variance were removed from further analysis.

**2.5.2    The coefficient of quartile variation (CQV)**
We used quantile based statistical metrics for descriptive statistics and correlation since throughfall and soil
moisture patterns are commonly skewed (Famiglietti et al., 1998; Zimmermann and Zimmermann, 2014), and
throughfall typically includes extreme values due to dripping points (Falkengren-Grerup, 1989; Keim et al., 2005;
Staelens et al., 2006; Voss et al., 2016). For the coefficient of variation, we used the quartile variation coefficient
(CQV) (Bonett, 2006) as alternative to the coefficient of variation:

$$CQV = \frac{Q3 - Q1}{Q3 + Q1}$$

where Q1 and Q3 represent first and third quartiles. Like the classical coefficient of variation, the CQV is
dimensionless statistical measure that describes the relative degree of scattering of the sample.

**2.5.3    Linear mixed effects models calculation**
We applied linear mixed effect models (LME) with repeat-measurement structure to evaluate the influence of
potential drivers explaining soil water content or soil water content increase. We present results on the following
dependent variables: Spatial pattern of pre-event ($\delta\theta_{pre}$), and post-event ($\delta\theta_{post}$) soil water content as well as soil
water content increase ($\Delta\theta$).
The independent variables (fixed effects) for $\delta\theta_{pre}$ were: Gross precipitation ($P_g$), nearest tree distance ($d_{tree}$), air
capacity ($\theta_{AC}$), field capacity ($\theta_{FC}$), throughfall of the preceding event ($P_{TFpre}$). The independent variables (fixed
effects) for $\Delta\theta$ and $\delta\theta_{post}$ were: Gross precipitation ($P_g$), spatial median of soil pre-event water content ($\hat{\theta}_{pre}$),
spatial pattern of soil pre-event water content ($\delta\theta_{pre}$), nearest tree distance ($d_{tree}$), air capacity ($\theta_{AC}$), field capacity
($\theta_{FC}$), spatial median of throughfall ($\hat{P}_{TF}$) and spatial pattern of throughfall ($\delta P_{TF}$). Year, day of year and sensor
position were implemented as random effects accounting for repeated measurements. To avoid model over-fitting
it is important that there are no strong correlations between the explanatory variables (Graham, 2003). To detect
multi-collinearity and to avoid potentially spurious models we calculated Spearman rank correlation coefficients
($\rho$) for all pairs of predictors (Table S1). Before the analysis we removed one of a pair of highly correlated
predictors: Gross precipitation ($P_g$, strong correlation with $\hat{P}_{TF}$) and field capacity ($\theta_{FC}$, strong correlation with
$\theta_{AC}$). All predictor variables were normalized. To obtain the minimal adequate models for the response variables,





we started with the maximum model and removed stepwise all non-significant terms based on the Akaike
Information Criterion (AIC). Main effects included in significant interactions were retained in the model.

## 3. Results

### 3.1 Precipitation, throughfall and soil water content pattern

The summer rainfall (May to October) for the last 30 years (1986 – 2016) shows an average of 352 mm
(Mühlhausen-Windeberg). During the two summer periods of this study (2015 and 2016), the annual rainfall was
below the long-term mean (276 and 303 mm, respectively). However, the summer 2015 were the third driest of
the last 30 years (Metzger et al., 2017). The final winter months of 2014 were the driest and the hydrological year
2014/2015 the second driest of the 30 years period. The hydrological year 2015/2016 and the final winter months
of 2015 received average precipitation.
Descriptive statistics of throughfall and soil water content (topsoil and subsoil) are given in Table 1. We observed
14 rainfall events in 2015 and ten in 2016. The gross precipitation ranged between 1.6 and 35.2 mm, with three
small, six medium and five large in 2015, and one medium and nine large events in 2016. For both years, soil
water content increased with soil depth (Table 1). The soil water content increase (difference between post-event
and pre-event soil water content; $\Delta\theta$) was always higher in the topsoil compared to the subsoil. For smaller rainfall
events, an increase in soil water content was mainly limited to the topsoil, and only following larger rainfall event,
in both soil depths.

### 3.2 Spatial pattern of throughfall

The model parameters fitted to the semi-variograms in the separate steps indicated in Section 2.5.1 are shown in
Table S2-4 and correlation lengths (effective range) of the final variograms (step 4, Table S4) are shown in Figure
2. Throughfall correlation lengths decreased with increasing event size from on average 6.2 m for large events to
7.5 m for medium and 9.5 m for small events. In comparison, canopy density correlation length was 7.5 m, i.e.
similar to medium events. Throughfall and canopy density had a small nugget and a strong spatial dependence
(nugget/sill ratio < 25%) for all events (Table S4). For both years, throughfall decreased significantly with
increasing canopy density (Table S5), although most of the variance for spatial patterns of throughfall was related
to unknown random effects.
The spatial variation of throughfall (inter-quartile range) increased with event throughfall, but the coefficient of
quartile variation (CQV), which normalizes by event size, decreased (Table 1). The high Spearman rank correlation
coefficient indicates a strong similarity of the spatial distribution of throughfall between individual events of the
same size class (Figure 3). Thus, throughfall produced persistent wet and dry spots, also confirmed by time stability
plots (Figure S2).
Soil water content spatial variation coefficients (CQV) decreased with increasing soil water content (expressed as
the spatial mean) and consequently with increasing soil depth (Table 1, Figure S3). In the topsoil, the relation was
more concave for post-event soil water content (Figure S3) compared to pre-event soil water content, indicating
that the event response enhanced soil water content variation especially in drier (summer) conditions in topsoil.
However, the by far highest CQV were observed for the increase in soil water content after rain ($\Delta\theta$).



**Table 1:** Overview of observed rainfall event properties. Event date, gross precipitation ($P_g$), spatial statistics of throughfall ($P_{TF}$), soil water content before ($\theta_{pre}$) and after ($\theta_{post}$) the rain event, as well as the soil water content increase ($\Delta\theta$) in topsoil and subsoil: spatial median (med), coefficient of quartile variation (CQV), interquartile range (IQR), and effective range (Range).

| Date | Precipitation | | | | | | Topsoil water content | | | | | | Subsoil water content | | | | | |
|---|---|---|---|---|---|---|---|---|---|---|---|---|---|---|---|---|---|---|
| | $P_G$ | Event size | $P_{TF}$ | | | | $\theta_{pre}$ | | $\theta_{post}$ | | $\Delta\theta$ | | $\theta_{pre}$ | | $\theta_{post}$ | | $\Delta\theta$ | |
| | | | med | CQV | IQR | Range | med | CQV | med | CQV | med | CQV | med | CQV | med | CQV | med | CQV |
| | mm | | mm | - | mm | m | Vol-% | - | Vol-% | - | Vol-% | - | Vol-% | - | Vol-% | - | Vol-% | - |
| 21.07.2015 | 1.6 | small | 0.6 | 0.29 | 0.4 | 9.6 | 21 | 0.16 | 21 | 0.17 | 0.08 | 2.6 | 36 | 0.10 | 36 | 0.10 | -0.04 | -3.34 |
| 20.06.2015 | 2.1 | small | 0.4 | 0.60 | 0.5 | 9.8 | 19 | 0.15 | 19 | 0.15 | 0.00 | 5.0 | 30 | 0.13 | 30 | 0.13 | 0.30 | 0.27 |
| 30.05.2015 | 2.8 | small | 1.7 | 0.21 | 0.7 | 9.2 | 27 | 0.14 | 27 | 0.14 | 0.03 | 1.0 | 37 | 0.11 | 38 | 0.11 | 0.00 | -1.00 |
| 18.06.2015 | 3.3 | medium | 1.8 | 0.28 | 1.0 | 5.8 | 19 | 0.15 | 20 | 0.16 | 0.03 | 1.0 | 31 | 0.13 | 31 | 0.13 | 0.00 | -1.47 |
| 13.07.2015 | 3.3 | medium | 1.9 | 0.22 | 0.8 | 8.6 | 17 | 0.14 | 17 | 0.14 | -0.02 | 41.0 | 27 | 0.14 | 27 | 0.15 | -0.01 | - |
| 02.06.2015 | 3.7 | medium | 1.8 | 0.25 | 0.9 | 8.0 | 27 | 0.15 | 27 | 0.15 | 0.00 | 3.0 | 37 | 0.12 | 37 | 0.12 | 0.00 | - |
| 13.05.2015 | 4.1 | medium | 2.7 | 0.19 | 1.0 | 7.6 | 34 | 0.11 | 35 | 0.10 | 0.71 | 0.89 | 41 | 0.08 | 41 | 0.08 | -0.01 | -1.00 |
| 11.07.2015 | 4.6 | medium | 2.7 | 0.13 | 0.7 | 8.9 | 17 | 0.14 | 18 | 0.13 | 0.13 | 1.00 | 27 | 0.14 | 28 | 0.14 | 0.72 | 0.32 |
| 25.07.2015 | 5.7 | medium | 3.9 | 0.14 | 11 | 4.6 | 19 | 0.13 | 21 | 0.14 | 0.41 | 0.98 | 33 | 0.11 | 33 | 0.11 | 0.00 | -3.00 |
| 15.07.2015 | 10.5 | large | 6.6 | 0.18 | 2.4 | 5.9 | 17 | 0.14 | 19 | 0.17 | 1.5 | 0.76 | 27 | 0.14 | 28 | 0.14 | 0.33 | 0.65 |
| 08.07.2015 | 13.3 | large | 9.4 | 0.08 | 1.50 | 4.8 | 17 | 0.14 | 19 | 0.15 | 2.0 | 0.78 | 28 | 0.13 | 29 | 0.13 | 0.28 | 0.87 |
| 28.07.2015 | 20.1 | large | 13.7 | 0.16 | 4.4 | 7.5 | 19 | 0.13 | 23 | 0.21 | 4.1 | 0.57 | 32 | 0.12 | 35 | 0.12 | 2.60 | 0.71 |
| 24.06.2015 | 23.0 | large | 14.2 | 0.15 | 4.4 | 7.0 | 19 | 0.15 | 24 | 0.21 | 5.2 | 0.66 | 30 | 0.13 | 31 | 0.13 | 0.27 | 0.86 |
| 20.07.2015 | 35.2 | large | 29.2 | 0.06 | 3.5 | 5.9 | 16 | 0.15 | 22 | 0.19 | 6.4 | 0.56 | 27 | 0.14 | 33 | 0.14 | 5.43 | 0.65 |
| 28.06.2016 | 5.3 | medium | 2.6 | 0.25 | 1.3 | 7.8 | 26 | 0.13 | 25 | 0.14 | 0.00 | -1.00 | 35 | 0.11 | 35 | 0.11 | 0.00 | -1.00 |
| 21.06.2016 | 13.7 | large | 10.1 | 0.13 | 2.6 | 8.9 | 34 | 0.10 | 38 | 0.09 | 3.90 | 0.23 | 39 | 0.09 | 42 | 0.09 | 1.56 | 0.53 |
| 06.06.2016 | 16.9 | large | 14.9 | 0.09 | 2.8 | 3.0 | 34 | 0.09 | 39 | 0.09 | 4.33 | 0.31 | 41 | 0.09 | 43 | 0.08 | 1.58 | 0.43 |
| 02.08.2016 | 19.6 | large | 13.7 | 0.11 | 3.1 | 5.7 | 20 | 0.13 | 22 | 0.19 | 2.17 | 0.81 | 30 | 0.13 | 31 | 0.13 | 0.12 | 0.99 |
| 04.07.2016 | 19.8 | large | 11.9 | 0.14 | 3.4 | 9.5 | 23 | 0.14 | 25 | 0.16 | 1.60 | 0.83 | 32 | 0.11 | 33 | 0.11 | 0.01 | 1.51 |
| 25.05.2016 | 20.8 | large | 13.3 | 0.11 | 3.1 | 6.5 | 26 | 0.12 | 33 | 0.15 | 5.77 | 0.50 | 37 | 0.11 | 39 | 0.11 | 0.74 | 0.96 |
| 16.06.2016 | 23.2 | large | 15.2 | 0.11 | 3.3 | 7.3 | 35 | 0.12 | 37 | 0.10 | 2.21 | 0.27 | 40 | 0.09 | 40 | 0.09 | 0.01 | 5.84 |
| 14.07.2016 | 24.1 | large | 20.0 | 0.10 | 4.0 | 5.0 | 22 | 0.17 | 23 | 0.20 | 0.99 | 0.89 | 39 | 0.09 | 42 | 0.09 | 2.81 | 0.50 |
| 31.05.2016 | 25.0 | large | 21.0 | 0.11 | 4.4 | 4.6 | 30 | 0.12 | 39 | 0.09 | 8.05 | 0.21 | 39 | 0.09 | 43 | 0.09 | 3.98 | 0.38 |
| 25.07.2016 | 33.5 | large | 25.6 | 0.13 | 6.6 | 3.5 | 22 | 0.15 | 23 | 0.18 | 0.42 | 0.96 | 33 | 0.13 | 35 | 0.13 | 1.34 | 0.48 |
| | 2.2 | small | 0.9 | 0.4 | 0.54 | 9.5 | 22 | 0.15 | 23 | 0.15 | 0.04 | 2.87 | 34 | 0.11 | 35 | 0.11 | 0.09 | -1.36 |
| | 4.3 | medium | 2.5 | 0.2 | 0.95 | 7.3 | 23 | 0.15 | 23 | 0.15 | 0.2 | 6.67 | 33 | 0.11 | 33 | 0.11 | 0.11 | -1.23 |
| | 20.3 | large | 14.8 | 0.1 | 3.54 | 5.6 | 23 | 0.13 | 27 | 0.13 | 3.27 | 0.62 | 34 | 0.11 | 36 | 0.11 | 1.40 | 0.82 |





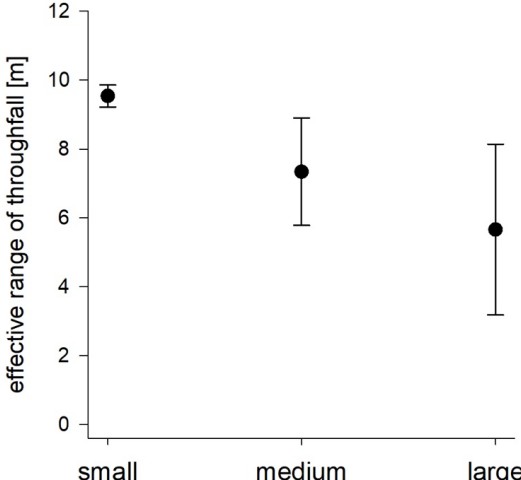

Fig. 2: Comparison of the correlation length, given as effective range, derived from the throughfall variogram
calculated for small ($P_g < 3$ mm), medium (3 mm $< P_g <$ 10 mm), large ($P_g > 10$ mm) events.

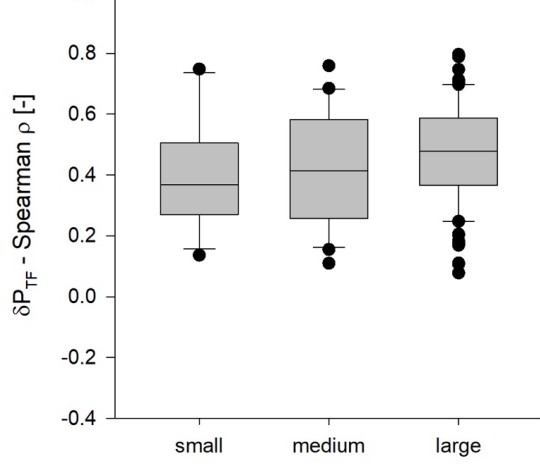

Fig. 3: Temporal stability of the spatial throughfall patterns. Shown are the pairwise correlation coefficients
(Spearman) between throughfall (normalized deviation from the plot median ($\delta P_{TF}$)) from different precipitation
events, grouped by event size class (small (n=11), medium (n=21), large (n=91) events.





The pairwise correlation coefficients indicating the temporal stability of the spatial patterns were high for pre-
event (drained) soil water content ($\theta_{pre}$) both in topsoil (Figure 4a) and subsoil (Figure 4b) with $\rho \approx 0.78$. For post-
event soil water content ($\theta_{post}$) they were significantly lower in the topsoil ($\rho = 0.70$, Figure 4c) than subsoil
($\rho = 0.77$, Figure 4d) (Mann-Whitney-U Test: $Z = -3.15$, $p = 0.002$). In the topsoil they decreased with increasing
event size, revealing patterns were less similar after large precipitation events (Figure 4a,c). In contrast, patterns
in soil water content increase after rain events ($\Delta\theta$) were much more weakly correlated with each other (Figure
4e,f). However, the similarity of the patterns increased with event size especially in topsoil (Figure 4e), confirming
reoccurring wetting patterns especially following larger events.

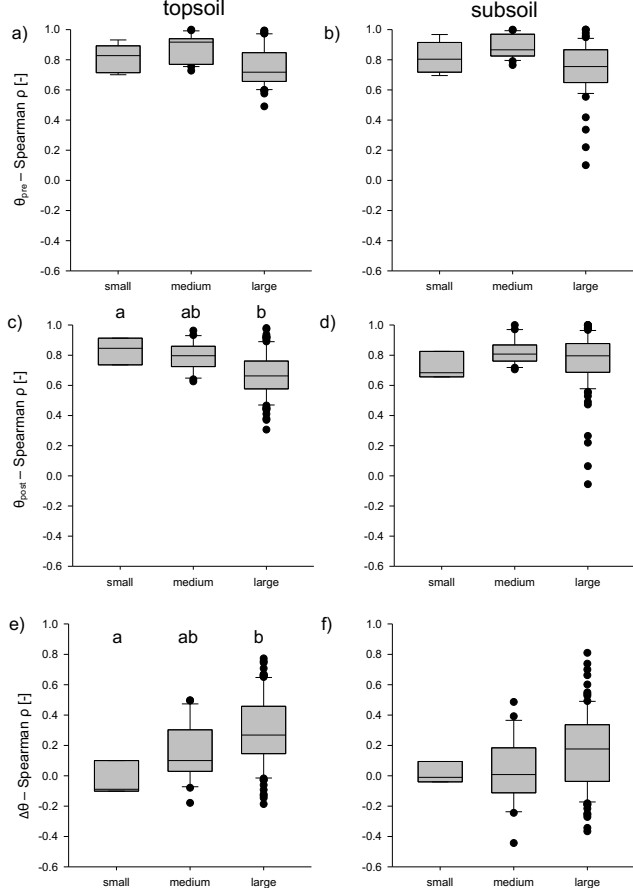

**Fig. 4**: Temporal autocorrelation of spatial patterns of pre- and post-event soil water content and increase of soil
water content after rainfall calculated as pairwise correlation coefficients (Spearman $\rho$) between all of the
different precipitation events within the event size class (small (n = 3), medium (n = 21), large (n = 91)). **(top)**
pre-event soil water content ($\theta_{pre}$); **(middle)** post-event soil water content ($\theta_{post}$); **(bottom)** increase of soil water
content ($\Delta\theta_i$); **(left)** topsoil; **(right)** subsoil. The differences between the events were examined using the Duncan
post hoc test of a one-way ANOVA. Letters on the top of bars indicate significant difference ($p \leq 0.05$) between
the groups.



### 3.3 Factors influencing soil water spatial distribution

#### 3.3.1 Soil water content

In order to identify the basic drivers for the patterns of soil water content in the drained state ($\delta\theta_{pre}$), we used mixed effects model selection. The resulting best models for top- and subsoil are given in Table 2. The variance explained by fixed effects (marginal $R^2$) was low, whereas the variance explained by fixed and random effects together (conditional $R^2$) was high. The model for the subsoil showed an even higher marginal $R^2$ compared to the topsoil, and a somewhat higher influence of fixed effects. The most important effect identified for topsoil and subsoil was air capacity, with lower soil water content ($\delta\theta_{pre}$) related to locations of higher air capacity (Table 2). In the topsoil also the throughfall of the preceding precipitation event slightly affected the soil moisture pattern. The results for the soil water content itself in the drained state ($\theta_{pre}$) are similar to those of $\delta\theta_{pre}$, except that fixed effects explain even less variation (Table S6).

**Table 2:** Factors affecting pre-event soil water content patterns ($\delta\theta_{pre}$) in topsoil and subsoil. Results for the best linear mixed effects model. Significant effects are highlighted in bold.

|  | topsoil | | subsoil | |
|---|---|---|---|---|
| *Explained variation* | | | | |
| R² Full model | 0.818 | | 0.822 | |
| R²  Fixed | 0.035 | | 0.143 | |
| R²  Random | 0.783 | | 0.679 | |
|  | t-value | p-value | t-value | p-value |
| *Fixed effects* | | | | |
| ***Air capacity, $\theta_{AC}$*** | -2.5 | ***0.013*** | -3.7 | ***<0.001*** |
| ***Throughfall of previous event, $P_{TF, prev}$*** | 2.3 | ***0.039*** | -1.5 | 0.161 |
| Tree distance, $d_{tree}$ | -0.8 | 0.426 | 1.8 | *0.065* |
| *Interactions* | | | | |
| ***$P_{TF,prev}$ x $\theta_{AC}$*** | - | - | -2.7 | ***0.007*** |
| ***$P_{TF, prev}$ x $d_{tree}$*** | -2.0 | ***0.047*** | - | - |
| $\theta_{AC}$ x $d_{tree}$ | - | - | 1.9 | *0.057* |

The results of the best linear mixed effects model relating soil water content after a precipitation event to potential drivers is given in Table 3.  The spatial pattern of soil water content before the rain event ($\delta\theta_{pre}$) was the major control on either absolute values of spatially distributed soil water content after the rain event ($\theta_{post}$, Table 3) or its spatial pattern ($\delta\theta_{post}$, Table S8). Other fixed ($\hat{P}_{TF}, \delta P_{TF}, \hat{\theta}_{pre}, \theta_{AC}, d_{tree}$ ) and random effects explained only a very small part of the variation.



**Table 3:** Factors influencing soil water content after a precipitation event ($\theta_{post}$). Results for the best linear mixed effects model including all data (left columns) and grouped by event size (small, medium and large, right columns). Significant effects are shown in bold and effects that were significant in both soil depth (based on all events) are highlighted in grey. Variables are scaled such that the t-value indicates the effect strength. Pseudo R² values are given separately for fixed and random effects.

| | topsoil | | | | | | | | subsoil | | | | | | | |
| --- | --- | --- | --- | --- | --- | --- | --- | --- | --- | --- | --- | --- | --- | --- | --- | --- |
| | All events | | Small events | | Medium events | | Large event | | All events | | Small events | | Medium events | | Large events | |
| | t-value | p-value | t-value | p-value | t-value | p-value | t-value | p-value | t-value | p-value | t-value | p-value | t-value | p-value | t-value | p-value |
| Full model R² | 0.90 | | 0.99 | | 0.96 | | 0.83 | | 0.89 | | 0.86 | | 0.92 | | 0.92 | |
| Fixed effects R² | 0.87 | | 0.99 | | 0.96 | | 0.76 | | 0.88 | | 0.86 | | 0.92 | | 0.91 | |
| Random effects R² | 0.03 | | 0.00 | | 0.00 | | 0.07 | | 0.00 | | 0.00 | | 0.00 | | 0.01 | |
| *Fixed effects* | | | | | | | | | | | | | | | | |
| Median event throughfall, $\hat{P}_{TF}$ | 2.2 | **0.035** | 4.1 | **<0.001** | 2.8 | **0.007** | 1.2 | 0.238 | -0.6 | 0.494 | - | - | -3.0 | **0.003** | -1.6 | 0.141 |
| Spatial throughfall pattern, $\delta P_{TF,i}$ | 1.8 | 0.072 | - | - | 2.0 | 0.051 | 1.9 | 0.062 | 3.0 | **0.003** | - | - | 0.8 | 0.408 | 3.0 | **0.003** |
| Initial median soil water content, $\theta_{pre}$ | -1.7 | 0.106 | -4.0 | **<0.001** | -2.3 | **0.038** | -2.5 | **0.024** | -1.4 | 0.184 | - | - | -1.2 | 0.211 | 0.6 | 0.582 |
| Spatial pattern of initial soil water content, $\delta\theta_{pre,i}$ | 76.4 | **<0.001** | 200.0 | **<0.001** | 97.2 | **<0.001** | 39.2 | **<0.001** | 98.4 | **<0.001** | 41.4 | **<0.001** | 79.2 | **<0.001** | 59.7 | **<0.001** |
| Tree Distance, $d_{tree}$ | 1.8 | 0.081 | - | - | - | - | 0.9 | 0.394 | - | - | - | - | - | - | 0.7 | 0.498 |
| Air capacity, $\theta_{AC,i}$ | - | - | - | - | -2.5 | **0.013** | - | - | -2.1 | **0.042** | - | - | - | - | -2.2 | **0.035** |
| *Interactions* | | | | | | | | | | | | | | | | |
| $\hat{P}_{TF}$ x $\delta P_{TF,i}$ | 2.3 | **0.019** | - | - | - | - | - | **0.015** | 2.0 | **0.048** | - | - | 2.0 | **0.043** | - | - |
| $\hat{P}_{TF}$ x $d_{tree,i}$ | -1.5 | 0.129 | - | - | - | - | -2.4 | **0.015** | - | - | - | - | - | - | -2.2 | **0.026** |
| $\hat{P}_{TF}$ x $\theta_{pre}$ | - | - | - | - | - | - | - | - | - | - | - | - | - | - | - | - |
| $\hat{P}_{TF}$ x $\delta\theta_{pre,i}$ | -9.7 | **<0.001** | -2.5 | **0.011** | - | - | -2.2 | **0.027** | -2.1 | **0.032** | - | - | -5.7 | **<0.001** | -2.6 | **0.010** |
| $\hat{P}_{TF}$ x $\theta_{AC,i}$ | - | - | - | - | - | - | - | - | - | - | - | - | - | - | - | - |
| $\theta_{AC,i}$ x $\delta P_{TF,i}$ | - | - | - | - | - | - | - | - | -3.4 | **<0.001** | - | - | - | - | -3.7 | **<0.001** |
| $\theta_{AC,i}$ x $d_{tree}$ | - | - | - | - | 2.3 | **0.021** | - | - | - | - | - | - | - | - | - | - |
| $\theta_{AC,i}$ x $\delta\theta_{pre,i}$ | - | - | - | - | - | - | - | - | 2.9 | **0.003** | - | - | - | - | 3.6 | **<0.001** |
| $\hat{\theta}_{pre}$ x $\delta P_{TF,i}$ | -2.0 | **0.019** | - | - | - | - | - | - | -2.8 | **0.004** | - | - | - | - | - | - |
| $\theta_{pre}$ x $d_{tree}$ | - | - | - | - | 2.2 | **0.025** | - | - | - | - | - | - | - | - | - | - |
| $\theta_{pre}$ x $\delta\theta_{pre,i}$ | 3.2 | **0.002** | - | - | - | - | 3.5 | **<0.001** | - | - | - | - | 1.7 | 0.082 | - | - |
| $\delta\theta_{pre,i}$ x $\delta P_{TF,i}$ | -2.0 | **0.019** | - | - | - | - | - | - | -2.8 | **0.004** | - | - | - | - | -3.6 | **<0.001** |





### 3.3.2 Soil water response (Δθ)

The models for explaining the soil water content increase (Δθ), i.e. how much water was locally stored in the soil after rain, are shown in Table 4. In general, a detectable (> 1%) change of $\Delta\theta_i$ was limited to large rainfall events (Table 1). The spatial patterns responded to several drivers (fixed effects) in the final model. There, the variance explained by fixed effects (marginal $R^2$) was generally higher for subsoil compared to topsoil, it typically increased with event size and was highest for the models including all event sizes (Table 4). In the following we therefore focus on the effects emerging from those latter models, that is the ones including all events, while the results for the individual event size classes are used only for more detailed interpretation. The grey shaded lines highlight the significant relations that occurred both in top- and subsoil.

Overall, local soil water content increase (Δθ) depended not only on event median throughfall ($\hat{P}_{TF}$), but also on the spatial pattern of throughfall ($\delta P_{TF}$) and spatial patterns of initial or pre-event soil moisture ($\delta\theta_{pre}$). Nearly all main effects are also included in interactions, meaning that likely a third variable influenced the relationship between an independent and dependent variable. For example, locally elevated throughfall enhanced the soil water increase (Table 4), but more so with increasing event size (interaction $\hat{P}_{TF}$ x $\delta P_{TF}$, visualized in Figure 5 a and b).

Spatial patterns of pre-event (or initial or drained) soil water content ($\delta\theta_{pre}$) notably affected top- and subsoil differently, making it the only factor yielding opposite effects on soil water content increase in different soil depths. In topsoil, drier locations stored less water per event than moister spots (positive t-value), whereas in subsoil, the opposite was the case (negative t-value). The influence of pre-event soil moisture patterns increased with event size (interaction $\hat{P}_{TF}$ x $\delta\theta_{pre,}$). Note that the slope of the interaction (represented by the sign of the t-value) changes with overall soil water conditions consistently in both depths (Table 4, interaction $\hat{\theta}_{pre}$ x $\delta\theta_{pre}$, visualized in Figure 6a): Locally drier soil increased soil water storage in wet, but decreased it in dry times.

Additional factors affecting the soil water response in top soil were related to the distance to the next tree. Locations near trees reacted stronger to event precipitation than those further away (interactions $\hat{P}_{TF}$ x $d_{tree}$), but only in overall moister soil conditions (Table 4, interaction $\hat{\theta}_{pre}$ x $d_{tree}$). In the subsoil higher air capacity ($\theta_{AC}$), representing the higher macropore volume, dampened the soil water response (Table 4, negative t-value), and more so when or where throughfall was high (interactions $\hat{P}_{TF}$ x $\theta_{AC}$ and $\theta_{AC}$ x $\delta P_{TF}$) as well as in drier locations (interaction $\theta_{AC}$ x $\delta\theta_{pre}$).





**Table 4:** Factors influencing local soil water content response after rainfall ($\Delta\theta_i$, i.e. difference between soil water content after and before each event). Results for the best linear mixed effects model including all data (left columns), and grouped according to event size (small, medium and large, right columns). Significant effects are shown in bold and factors with significant effects in both depth (based on all all events) are highlighted in grey. Variables are scaled such that the t-value indicates the effect strength. Pseudo R² values are given separated for fixed and random effects.

| | topsoil | | | | | | | | subsoil | | | | | | | |
| --- | --- | --- | --- | --- | --- | --- | --- | --- | --- | --- | --- | --- | --- | --- | --- | --- |
| | All events | | Small events | | Medium events | | Large event | | All events | | Small events | | Medium events | | Large events | |
| | 0.57 | | 0.10 | | 0.32 | | 0.54 | | 0.62 | | 0.38 | | 0.46 | | 0.55 | |
| Full model R² | 0.25 | | 0.09 | | 0.12 | | 0.10 | | 0.38 | | 0.38 | | 0.04 | | 0.27 | |
| Fixed effects R² | 0.32 | | 0.01 | | 0.20 | | 0.43 | | 0.24 | | 0.00 | | 0.42 | | 0.28 | |
| Random effects R² | t-value | p-value | t-value | p-value | t-value | p-value | t-value | p-value | t-value | p-value | t-value | p-value | t-value | p-value | t-value | p-value |
| **_Fixed effects_** | | | | | | | | | | | | | | | | |
| Median event throughfall, $\bar{P}_{TF}$ | **5.1** | **<0.001** | - | - | **3.5** | **0.014** | 1.3 | 0.215 | **6.8** | **<0.001** | **3.4** | **<0.001** | - | - | **4.1** | **0.001** |
| Spatial throughfall pattern, $\delta P_{TF,i}$ | **2.0** | **0.044** | - | - | **2.2** | **0.030** | **2.2** | **0.031** | **3.0** | **0.003** | - | - | - | - | **3.3** | **<0.001** |
| Initial median soil water content, $\bar{\theta}_{pre}$ | 0.4 | 0.712 | **5.6** | **<0.001** | 1.3 | 0.241 | 0.9 | 0.382 | -0.3 | 0.787 | **-10.9** | **<0.001** | - | - | 0.2 | 0.843 |
| Spatial pattern of initial soil water content, $\delta\theta_{pre,i}$ | **4.2** | **<0.001** | - | - | - | - | **3.3** | **0.001** | **-3.9** | **<0.001** | **-2.4** | **0.017** | - | - | **-4.3** | **<0.001** |
| Tree Distance, $d_{tree}$ | 1.3 | 0.211 | - | - | **2.9** | **0.004** | 0.6 | 0.569 | - | - | - | - | 1.5 | 0.144 | - | - |
| Air capacity, $\theta_{AC,i}$ | - | - | - | - | -0.8 | 0.398 | - | - | **-2.4** | **0.016** | -1.4 | 0.173 | -1.0 | 0.337 | **-2.8** | **0.007** |
| **_Interactions_** | | | | | | | | | | | | | | | | |
| $\bar{P}_{TF} \times \delta P_{TF,i}$ | **2.4** | **0.016** | - | - | - | - | - | - | **3.7** | **<0.001** | - | - | - | - | - | - |
| $\bar{P}_{TF} \times d_{tree,i}$ | **-2.3** | **0.020** | - | - | - | - | **-2.8** | **0.005** | - | - | - | - | **2.5** | **0.012** | - | - |
| $\bar{P}_{TF} \times \bar{\theta}_{pre}$ | - | - | - | - | - | - | - | - | - | - | - | - | - | - | - | - |
| $\bar{P}_{TF} \times \delta\theta_{pre,i}$ | **4.5** | **<0.001** | - | - | - | - | - | - | **-5.3** | **<0.001** | **3.3** | **<0.001** | - | - | **-3.3** | **<0.001** |
| $\bar{P}_{TF} \times \theta_{AC,i}$ | - | - | - | - | **-2.9** | **0.003** | - | - | **-3.3** | **0.001** | **2.6** | **0.011** | **2.6** | **0.009** | - | - |
| $\theta_{AC,i} \times \delta P_{TF,i}$ | - | - | - | - | - | - | - | - | **-4.0** | **<0.001** | - | - | - | - | **-3.8** | **<0.001** |
| $\theta_{AC,i} \times d_{tree}$ | - | - | - | - | **-2.6** | **0.012** | - | - | - | - | - | - | 1.9 | 0.061 | - | - |
| $\theta_{AC,i} \times \delta\theta_{pre,i}$ | - | - | - | - | - | - | - | - | **3.7** | **<0.001** | - | - | - | - | **3.5** | **<0.001** |
| $\bar{\theta}_{pre} \times \delta P_{TF,i}$ | - | - | - | - | **2.1** | **0.034** | **-2.1** | **0.033** | - | - | - | - | - | - | - | - |
| $\bar{\theta}_{pre} \times d_{tree}$ | **2.1** | **0.032** | - | - | **2.4** | **0.018** | - | - | - | - | - | - | - | - | - | - |
| $\bar{\theta}_{pre,i} \times \delta\theta_{pre,i}$ | **-6.5** | **<0.001** | - | - | - | - | **-7.3** | **<0.001** | **-2.5** | **0.011** | **-2.3** | **0.020** | - | - | **-2.6** | **0.011** |
| $\delta\theta_{pre,i} \times \delta P_{TF,i}$ | - | - | - | - | - | - | - | - | **-2.4** | **0.016** | - | - | - | - | **-2.0** | **0.048** |

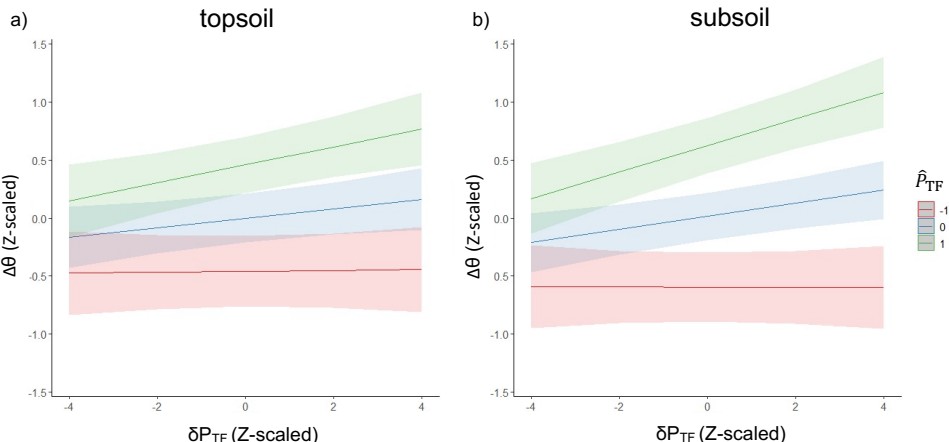


**Fig. 5**: Influence of the spatial pattern of throughfall ($\delta P_{TF}$) and on the soil water content response ($\Delta\theta$), grouped
by the event size (given as mean of throughfall, $\hat{P}_{TF}$) for (**left**) topsoil and (**right**) subsoil. Note that all values are
z-scaled, and therefore centered around zero. For example, the red line highlights events of below average
throughfall (small events). There, the spatial pattern of soil water content response depends little on that of
throughfall (small events). A stronger influence is seen for the above average (larger) events marked in green.

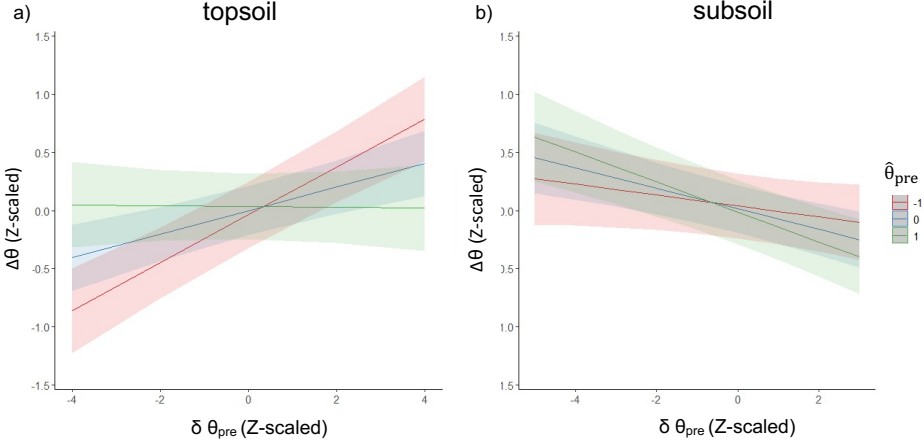


**Fig 6**. Interaction between the local soil water response to rain event ($\Delta\theta$) and the local pre-event soil water
content pattern ($\delta\theta_{pre}$) grouped by the pre-event spatial average soil moisture conditions (spatial median soil
water content, $\hat{\theta}_{pre}$) for (**left**) topsoil and (**right**) subsoil. For example, the green line shows that in overall moist
conditions (e.g. early spring), soil water content increase is dampened in moister locations (high values of $\delta\theta_{pre}$)
and more so on the subsoil. Dampening also takes place in drier locations in dry summer conditions in topsoil
(red line). Note: grouping according to soil water content ($\hat{\theta}_{pre}$) was done separately for topsoil and subsoil,
while absolute values in subsoil water content are always higher than in topsoil. Therefore, the shift in slopes
from positive to negative with soil moisture conditions is not only within but also between soil depth.



## 4. Discussion

### 4.1 Strengths and weaknesses of the approach

In this analysis we used extensive spatial data of canopy cover, throughfall and soil water content in order to assess the role of canopy processes on below-ground soil water response to precipitation. For this, we meassured precipitation and soil water content at different locations in order to avoid disturbance of soil water dynamics by the precipitation measurement and providing independent random measurement designs. To be able to relate observations at different locations, we used geostatistical methods to predict throughfall values at locations where soil water content was measured. Because the throughfall prediction can be based on an extensive dataset of 350 points, it allows reliable variogram estimations (Voss et al., 2016). Throughfall showed strong spatial autocorrelation which was reflected by nugget-to-still ratios much lower than 25% for all event sizes (Table S3). However, spatial correlation patterns depended on event size in that the correlation length decreased with increasing event rainfall. This decreased in larger events the range within which throughfall could be predicted and increased the number of locations with high kriging variance, that were removed from the analysis. As a result, this decreased the sample size for large compared to small and medium sized events. Regardless, for all sampled events, we could still rely on datasets of 59 points on average. Additionally, kriging predictions tend to be smoother compared to the actual data. However, the predicted values show approximately the same median and spatial variance as the measured data, indicating that the real variation was still maintained after the prediction procedure. Unfortunately, there is no perfect way to relate measurements obtained at different locations to each other. However, the combination of a large sample size of throughfall and variogram estimation by residual maximum likelihood (REML) seems to be a suitable way forward for interpolating the aboveground data to the belowground locations (Lark, 2000; Voss et al., 2016). Altogether, this provides a good basis to comparing above- and belowground measurements.

In our analysis we quantified only throughfall input and omit the role of stemflow, which may play a role in locations near stems. Extrapolating stemflow input to soil moisture locations entails more prediction steps compared to throughfall. Spatial variation of stemflow depends on the one hand on species, tree and canopy size, neighborhood and individual morphology of the trees (Bellot and Escarre, 1998; Fan et al., 2015b; Levia et al., 2014; Levia and Germer, 2015; Van Stan et al., 2016; Metzger et al., 2019; Magliano et al., 2019) and on the other hand on precipitation intensity and soil conditions determining the infiltration area (Herwitz, 1986; Carlyle-Moses et al., 2018; Metzger et al., 2021). Such a prediction would not only introduce a great deal of uncertainty, but also deviate from the main purpose of this study, which is to evaluate the role of throughfall heterogeneity. Therefore, in the model analysis, microsites near stems were accounted for by including distance to the stem as fixed effect in the model. This takes into account to some extent the potential influence of stemflow in the interpretation.

### 4.2 General patterns of throughfall (temporal and spatial)

In agreement with previous studies, throughfall patterns of large events show lower coefficients of variation compared to smaller ones (Aussenac, 1970; Loustau et al., 1992; Llorens et al., 1997; Su et al., 2019; Metzger et al., 2017; Carlyle-Moses, 2004; Staelens et al., 2008; Van Stan et al., 2020). Several other studies have suggested that throughfall spatial variation depend next to canopy characteristics also on precipitation amount (Loustau et al., 1992; Carlyle-Moses, 2004; Keim et al., 2005; Park and Cameron, 2008; Hsueh et al., 2016; Zimmermann et al., 2009). Similarly, at our site for all event size classes, canopy cover was a significant driver of throughfall spatial distribution, although a small one compared to the random effects. The correlation length (effective range) of throughfall decreased with increasing event size and corresponded for medium events roughly to that of canopy



cover. The change of spatial pattern with event size illustrates that not only canopy storage per se, but also other
processes like turbulence, wind shadows, the arrangement of canopy gaps, or the formation of canopy dripping
points can add persistent spatial organization to below-canopy precipitation (Carlyle-Moses, 2004; Keim et al.,
2005; Park and Cameron, 2008; Staelens et al., 2008; Zimmermann et al., 2008; Wullaert et al., 2009; Li et al.,
2016; Van Stan et al., 2020). In other words, not only canopy density, but also other canopy features probably
affect throughfall distribution (Park and Cameron, 2008; Zimmermann et al., 2009). Overall, and despite the slight
changes in throughfall correlation lengths for different events size classes, throughfall patterns were temporally
stable, indicating the existence of permanent hot and cold spots of throughfall, and those were consistent across
small, medium and large events. This is in line with several previous studies stating temporal stability of
throughfall patterns (Keim et al., 2005; Staelens et al., 2006; Wullaert et al., 2009; Zimmermann et al., 2009;
Fathizadeh et al., 2014; Fan et al., 2015b; Metzger et al., 2017; Molina et al., 2019; Zhu et al., 2021; Rodrigues et
al., 2022) even over several years (Wullaert et al., 2009; Rodrigues et al., 2022), although phenology and canopy
development have also been observed to deteriorate spatial stability (Zimmermann et al., 2008; Fathizadeh et al.,
2014). Furthermore, although spatial variation coefficients are smaller in large compared to small events, absolute
values vary much more in large events such that they have arguably a higher potential to induce spatial patterns in
soil water content or dynamics.

### 4.3 General soil water content patterns and potential drivers

Mean soil water contents were generally lower in the topsoil compared to the subsoil. At our site, the shallow soil
is underlain by undulating weathered calcareous bedrock (Kohlhepp et al., 2017) of low hydraulic conductivity,
and may locally be broken through by tree roots. While the topsoil is well-drained (i.e. saturated to field capacity
in winter and much lower in summer), the deeper and finer textured soil layer (Metzger et al., 2021) is influenced
by the much less conductive regolith and generally moister soil water content which very occasionally exceeds
field capacity in winter (Metzger et al., 2017).
Much in agreement with previous studies in humid regions (Brocca et al., 2007; Korres et al., 2015; Rosenbaum
et al., 2012; Metzger et al., 2017), spatial variation of soil water content increased in both top- and subsoil in drier
summer soil conditions. In an earlier study at the same site a strong but short-lived increase of spatial variation of
topsoil water content in summer was related to precipitation events (Metzger et al., 2017). Regardless, we found
that the main controlling factor of post-event soil water content was the spatial pattern of pre-event soil water
content, while average throughfall and spatial pattern of throughfall, tree distance and air capacity were additional,
but much less important drivers. In other words, while soil water content variation increases strongly after events,
this variation can only in very limited fashion be traced back to input patterns. This may in part be due to the small
inputs of water compared to the overall soil water storage, leading to a strong memory effect of the pre-event soil
water conditions on the post event patterns. Furthermore, preferential flow already taking place during the event
itself can blur the throughfall pattern within the soil storage (see below).
Soil water content spatial patterns in drained state in turn were strongly driven by random effects. Those are factors
that were not described by the measurements, but are temporally stable. Those so called local soil conditions are
potentially related to soil hydraulic properties, root water uptake and microtopography (Famiglietti et al., 1998;
Vereecken et al., 2007; Fan et al., 2015a). The mixed-effects models confirm, although with a very week influence,
that locations of higher air capacity (higher macroporosity) were drier in both depths, confirming the role of water
retention on soil water patterns (Metzger et al., 2017) at this site. Also, throughfall patterns of the previous event





slightly affected topsoil pre-event soil water content. Thus, an imprint of the throughfall pattern was carried over
to the next pre-event soil conditions, but this is barely detectable and negligible compared to the other sources of
variation in soil water content in drained state.

**4.4 Drivers of soil water response (Δ𝜃) to rainfall**
In contrast to the absolute values of soil water contents discussed above, the local soil water response (i.e. increase
of soil water content following rainfall events), was clearly driven by the spatial throughfall pattern both in top-
and subsoil. Since we tested the effect of the spatial pattern ($\delta P_{TF}$) separately from the absolute values of event
throughfall ($\hat{P}_{TF}$), we are able to demonstrate the influence of spatial throughfall specifically. Among all drivers
tested, the influence of spatial throughfall variation was the most consistent, appeared in both observed soil depths,
and was more pronounced for larger events. In other words, spatial patterns of throughfall were the most prominent
driver of soil wetting.
Measurements ascertaining that soil water content response relates to canopy drainage are comparatively rare.
Metzger et al. (2017) already reported for the same site, but a smaller dataset, that soil water content increase
correlated with event spatial throughfall patterns in larger rainfall events. Molina et al., (2019) found with highly
temporally resolved soil moisture measurements a weak relationship between the average pattern of throughfall
and that of soil water content response in the topsoil of a Mediterranean oak dominated forest plot, but not in a
pine plot. Notably, Klos et al. (2014) in a tropical rain forests showed that locations of high and low soil water
content below the main rooting zone corresponded to the end members of high and low throughfall, while soil
water content was more homogenous above and below this depth. They concluded from additional modelling that
preferential flow may have contributed to bypassing the main rooting zone. On the other hand, several studies,
such as Raat et al. (2002), Shachnovich and Berliner, (2008), and more recently Zhu et al. (2021) with temporally
less highly resolved soil water content measurements (incidentally all in coniferous forests) did not find relations
between the spatial patterns of soil water content and throughfall. All authors report that throughfall patterns were
pronounced and stable in time and suspect the forests floor hindered the transmission to soil water patterns. An
additional explanation could be that the effect of spatial net precipitation patterns on soil water content were so
short-lived (Metzger et al., 2017) due to preferential flow that they were not observed by infrequent hand
measurements. Altogether stronger soil water response at locations with above average throughfall indicates that
throughfall hot spots and also cold spots (Levia and Frost, 2006; Van Stan et al., 2020; Zimmermann et al., 2009)
translated into soil water dynamics, despite them going almost unnoticed in the soil water content pattern (see
above).
Next to the throughfall pattern, soil water response after large rainfall events depended in both depths also on the
pattern of pre-event soil water content. Notably, the slope of the relationship changes direction, making it the only
factor that shows opposite effects in the top- and subsoil. This can be attributed to its inter-dependence on soil
water content, and the difference in moisture between the two measurement depths. Especially in dry (summer)
conditions, wetter topsoil locations took up more of the arriving precipitation water, whereas drier locations
remained dry. This is a strong indication of preferential flow in dry soil, where e.g. hydrophobic conditions, cracks
and low hydraulic conductivity of the matrix can enhance preferential flow (Hillel, 1998; Nimmo, 2021; Beven
and Germann, 2013). On the other hand, the dampened water response in the wetter subsoil, could be due to
enhanced hydraulic conductivity and less free pore space (Vereecken et al., 2007; Hagen et al., 2020). Only in



intermediate soil water contents the spatial distribution of soil water contents had no influence on the spatial
drainage behavior.
Soil water response depended additionally also on the distance to the nearest tree in the topsoil and soil
properties (air capacity) in the subsoil. The enhanced moistening of soils near stems is likely related to stemflow
production (Metzger et al., 2019), which was not accounted for as input. Stemflow production generally
increases with event size (Levia and Germer, 2015; Metzger et al., 2019), explaining the interaction in the
model. The additional modification by soil water conditions can be explained by the systematically lower soil
water contents near tree trunks at the same site (Metzger et al., 2017, 2021), due to lower soil water retention and
likely enhanced drainage there.
Taken together, our data strongly suggest that additionally to spatial distribution of throughfall, the spatial pattern
in drainage behavior affects the local soil water response to rainfall. In that, both dry and wet locations can, water
supply permitting, act as percolation hotspots, depending on the overall soil conditions. Bypass flow in forests has
been repeatedly observed (e.g. Schume et al., 2003; Schwärzel et al., 2009; Bachmair et al., 2012; Blume et al.,
2009; Demand et al., 2019) especially in dry summer conditions (Schume et al., 2003; Bachmair et al., 2012;
Demand et al., 2019). Spatial variation of infiltration water supply and intensity, such as is the case for below
canopy precipitation (Keim and Link, 2018), has been suggested as a potential cause for initiating finger flow
(Nimmo, 2021), which is promoted by dry soil conditions. Also, hydrophobicity has been suggested to contribute
to maintaining recurring finger flow paths (Blume et al., 2009). Next to this, macropore flow along biopores
(Lange et al., 2009; Nespoulous et al., 2019) and soil cracks (Schume et al., 2003) can be enhanced in dry forest
soil conditions due to soil shrinking (Baram et al., 2012). While both finger flow and macropore flow may have
contributed to the observed patterns in soil water response, macropore flow more than finger flow could explain
enhanced matter export (Lehmann et al., 2021) as well as fast response following strong storms observed in the
shallow aquifers of the AquaDiva Critical Zone Observatory (Lehmann and Totsche, 2020).
Overall, our results confirm that the effect of throughfall on soil water content is weak, but stronger on the soil
water response. This contrasts with previous modelling (Coenders-Gerrits et al., 2013) that did not account for
preferential flow. With the effect of the throughfall pattern on the soil water response also depending on local
conditions related to hydraulic properties, its fate is much more likely to be found in the drainage fluxes, next to
the storage. The further destiny of the net precipitation pattern arguably depends on the deeper subsurface
hydrogeological setting. We deduce however, that net precipitation hotspots have a strong chance of producing
patterns of preferential flow below the main rooting zone, which is in line with previous work (Klos et al., 2014),
and backs earlier hypotheses (Bouten et al., 1992; Schume et al., 2003).

## 5. Conclusion

In this study, we collected an extensive dataset to investigate the effect of throughfall spatial heterogeneity on
the soil water response and checked which other factors (pre-event soil water content, macroporosity, tree
distance) modified the result. We first confirmed that throughfall patterns were stable in time and found that they
related to the vegetation canopy density, although additional and partly unknown factors strongly affected
throughfall distribution. We found that post event soil water content per se did have a very weak relation to
throughfall, although the variation of soil water content clearly increased in the aftermath of rain events. The
post-event soil water content pattern was overwhelmingly determined by the strong memory effect of the soil
water storage and only slightly affected by soil properties, like macroporosity. In contrast, the soil water
response showed a clear relation with the throughfall input pattern. In other words, our setup allowed us to



confirm experimentally that throughfall patterns do imprint on soil water content dynamics, at least shortly after
rain events. However, we also identified locations where soil water response was dampened, likely due to
enhanced fast drainage. Those locations could be either very dry locations likely promoting preferential flow,
especially in the topsoil, or wet locations, promoting faster release of the incoming water. Our results
demonstrate that throughfall spatial patterns leave a stronger imprint on soil water dynamics than on soil water
content directly, and explain why aboveground influence on soil hydrology has been so difficult to lay open in
the past. Our results are in line with previous research and contribute a more general process understanding of
the below ground consequences of precipitation redistribution by forests. Most importantly, our results strongly
suggest that throughfall patterns induce fast soil water flow with repeating spatial patterns. Those patterns would
therefore already be triggered within the canopy.

Acknowledgements
We thank Murray Lark for insightful comments on the strategy for comparing above and belowground spatial
patterns that helped shaping the geostatistical analysis.
This study is part of the Collaborative Research Centre AquaDiva of the Friedrich Schiller University Jena,
funded by the Deutsche Forschungsgemeinschaft (DFG, German Research Foundation) – SFB 1076 – Project
Number 218627073. We thank the Hainich CZE site manager Robert Lehmann and the Hainich National Park.
Field work permits were issued by the responsible state environmental offices of Thüringen.

Data availability
The dataset is currently prepared for publishing in a official repository. The doi will be posted with the data at
the latest when the data is published.

Author contributions
AH developed the project idea. All authors contributed to the collection of the raw data. CF conducted the
statistical analysis, developed it further with AH, and both wrote the first draft of the manuscript. All authors
contributed to writing of the manuscript.

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
