# Peer review of "Throughfall spatial patterns translate into spatial patterns of soil moisture dynamics – empirical evidence"

_Hydrology and Earth System Sciences, 2022_

## Author Response (AR1)

Dear Nadia Ursino, dear reviewers,

our thanks go to both reviewers for taking the time to reading the manuscript carefully, giving very constructive feedback, finding several errors and helping us improve the manuscript. We have responded to all comments in the point-by-point response below. In some very few instances, we have responded in a slightly different way, but hopefully in the sense intended by the reviewers.

Below the comments of the reviewers are printed in bold, the answer in normal fonts. Blue color highlights how we changed the manuscript. A version of the manuscript with changes highlighted in grey is attached.

Sincerely,

Anke Hildebrandt, Christine Fischer and co-authors.

**Response to Reviewer 1**

**[R1.1] This manuscript is an ambitious attempt to link spatial variability of throughfall with soil moisture, using an extremely detailed dataset and innovative throughfall interpolation technique. The results are explained well.**

Thank you very much for the positive feedback here and the very helpful comments below.

**[R1.2]The statistical treatment is relatively heavy, in that there are multiple normalizations, rank transformations, and per-event terms. Interpreting results is difficult as a result. Some additional discussion on this point would be helpful, as would more precision in the presentation of results. In general, the statistical treatment makes it impossible to determine how much difference throughfall variation makes for soil moisture. De-transforming after analysis or repeating the analysis with raw variables and generalized linear models that do not require pre-analysis transformations are two options that would make the results more useful.**

We agree. The statistical treatment is complex and difficult to follow. We worked on the communication in this version and give mode information in the point-by-point response below.

Regarding the specific concern on data transformation. The mixed effects models, used to test how spatial fields of throughfall affected soil moisture response, did not strictly require transformations. Instead, we transformed data for the following reasons:

1. We separated median and spatial deviation from the median in order to be able to distinguish between, e.g. precipitation event size and the spatial distribution of the throughfall. This allowed us to pool the data across events. The same is true for the soil water content. We explain this better now where this transformation is introduced.
2. We z-transformed the data in order to be able to compare the effect sizes, which are now also shown in the new Fig 5. It presents as proposed the slope estimates that give insight to how strongly e.g. throughfall affects soil water content increase. We added a motivation for this to methods section.
3. We think that with this additional explanation, both transformations carry more information than de-transformation in most situations. One exception is the grouping in Fig. 6 and 7, for which we now use de-transformed values.

**[R1.3] With a few mild assumptions, this manuscript appears to have enough data to do a full water balance of the soil, including deep percolation during events and transpiration between events. Those are probably for the future, but maybe they can be added easily here for a richer description of the importance of spatial variability of throughfall.**

Yes, there are several manuscripts in preparation dealing with the overall water balance of the forest plot, which turned out to be more complex than we expected. Therefore, the focus is really on different challenges. For example, determining transpiration from the soil water balance is not trivial in general (Hupet et al., 2003, Jackisch et al., 2020). One of the manuscripts in currently in discussion in HESS (Demir et al., in review), where we observed species interacting belowground. We sincerely think that separating those narratives helps focusing the message, and we therefore prefer to focus on the effect of throughfall on soil water response and do not extend to investigation of the soil water balance.

**Detailed comments:**

**[R1.4] L98 I suggest adopting "macroporosity" as it is here, instead of "air capacity" as most other places—that term is much less used.**

We changed the terminology from "air capacity" to "macroporosity" throughout the manuscript.

**[R1.5] L207 for how many events were there trends, and were those trends interpretable?**

Somewhat less than half of the events had spatial trends (Table S2) that we found difficult to interpret. They changed direction with time, were of varying strength and occurred in small as well as in large events. Canopy cover had no spatial trend (Table S2). Spatial trends may have been related to the interaction of slope aspect and wind direction. We added information and reference to the Table in the supplement in results section 3.2 (Lines 275-277), and a sentence to the discission section 4.2 (Lines 442-443).

**[R1.6] L218 citation should be Zimmerman et al. (2009). I found this sentence confusing and in need of explanation. I think it should say "we chose the estimator that gave median theta of 0.455" to match the box in Fig S1. If I understand this correctly, I think it means that you assumed the data were normally distributed and chose the estimator to match that, and that you feel confident in doing so because octile skew was never outside of [-2…2]. The manuscript would be better if that logic (or the correct logic) is described here. I also wonder what it means for the best estimator to vary. Is there something to infer about spatial structure of throughfall in each event?**

Thank you. We corrected reference.

We meant that we chose the variogram that yielded theta nearest to 0.455. We also reformulated the sentence accordingly (L 199-200).

Correct, we feel confident that the data is near normally distributed based on the octile skew test. This step 2 is meant to choose the right empirical variogram estimator and variogram model given that there may be spatial outliers present and that those differ from event to event. The procedure is based on Lark (2000) (cited in the manuscript), who proposed the $\Theta(x)$ as a statistic to support cross-validation of variograms specifically with keeping spatial outliers in mind:

$$\Theta(x) = \frac{\{z(x) - \tilde{Z}(x)\}}{\sigma_{K,x}^2}$$

with z(x) and $\tilde{Z}(x)$ the observed and kriged values of the variable z at the position x and $\sigma^2_{K,x}$ the kriging variance at the same location. In ideal conditions Θ(x) follows a $\chi^2$distribution. He argues that in the presence of spatial outliers, a comparison with the median of Θ(x) is more appropriate than using the mean, as the latter is not robust. That median is 0.455, if the correct variogram is used.

Yes, the spatial structure of the throughfall changes from event to event, which is partly due to the spatial outliers moving around, but also illustrated by the different correlation lengths and even spatial trends. Here we attempt to find the best possible statistical empirical and theoretical variogram to honor the individual events.

We think that the estimation procedure already takes a lot of space in the Methods section and we would therefore prefer to point the reader to the literature, where the details are laid out, while mainly guiding through the procedure here. We have reformulated this section (L 191-200) to improve navigation through the steps, while leaving out the details mentioned above.

**[R1.7] L225 how many outliers were removed, which events were they in, and is it interpretable as to why they are outliers? The answers to these questions would help readers understand the consequences of the removals.**

The number of spatial outliers changed substantially with event, and ranged between 3 and 16 points (1% to 4.5% of the of the observations). The average were 9 points (2.2% of the observations). For throughfall spatial outliers are typically expected at dripping points, where particularities in the canopy cause interruption of flow path along branches and locally strongly enhance throughfall. Here, only one third of the spatial outliers were related to dripping points. Although most (75%) of the spatial outliers corresponded to high throughfall locations exceeding the 0.9 quantile, only one third was also an outlier in the sample, exceeding the 0.99 quantile. As expected, most of the sample outliers naturally were also spatial outliers. The number of outliers was not related to the event size. We now discuss outlier removal in the discussion (Lines 410-415).

**[R1.8] L233 how exactly did you check? By whether theta was about 0.455?**

Yes, indeed. We edited the entire section to improve clarity (L191 – L200).

**[R1.9] L235 L444 why not use co-kriging, given the finding of correlation with local canopy density and distance to tree?**

Yes, based on the relation between canopy cover and throughfall co-kriging with canopy cover would theoretically be possible. However, unfortunately, the canopy cover was only assessed at the throughfall locations, and not at the soil sensor locations. We therefore cannot apply co-kriging to predict at the soil sensor locations, which was done here. There was no relation between throughfall and distance to the next tree (Table S4). A relation existed for soil water content, but not throughfall. Therefore, distance to the next tree could also not be used for co-kriging of throughfall to the soil sensor locations.

**[R1.10] L241 please resolve the conflict between this assertion of skew and previous assertion of no skew, necessary for adoption of gaussian assumptions in variogram model fitting.**

Good point. The octile skew of the throughfall was never beyond the threshold that required transformation. However, despite passing the octile skew test distributions are never ideal. Throughfall of small events can still be slightly skewed while medium ones are typically very well centered and especially large events often have some outliers. Therefore, we used robust estimators when comparing spatial distributions at the event scale. We also used quantile based robust

estimators in the variogram estimation (section 2.5.1, step 2) to identify spatial outliers. We included additional information to that section (Lines 221-224).

**[R1.10] L251 and elsewhere: naming delta-P, -TF, and -theta variables as "spatial pattern" is confusing because spatial pattern is a property of all locations, not of a single location. An example of how this term is problematic is L323: patterns cannot be correlated with each other, and there is really only one pattern for each time.**

We understand that a pattern is a property of the entire sample, and we realize that we have used the term interchangeably. We have revisited the mentions of the term "pattern" and checked for consistent use and adjusted wording to avoid confusion. However, in the specific sentence (starting Line 323 in the old and Line 311 in the new manuscript), we refer to the correlation between spatial samples of soil moisture response after different events. We believe the term is used properly. We state that those patterns were less correlated with each other than those of the soil water content. Maybe the parallel use of $\Delta\theta$ and $\delta\theta$ creates confusion. Unfortunately, both are very established symbols and we think changing would not necessarily yield the desired improvement in communication. We therefore now add at several occasions the information that $\Delta\theta = \theta_{post} \theta_{pre}$, to improve readability. We also include a new Table 1, summarizing the response and predictor variables as well as indices and operators used in the manuscript.

**[R1.1] L263 normalizing predictor variables is a very important decision that affects model selection and interpretation. Unfortunately, it is difficult to interpret here because "normalize" does not have a consistent meaning and the exact transformation is not specified (Fig 5 axis labels suggest z transform). Was the transformation parametric, and, if so, how well is that choice justified given the rank transformation used on many other variables, including most response variables? Why were all transformations necessary? Axis labels fig 5 and 6 suggest the response variables were also transformed, but that is not mentioned in the methods. Finally, the results and discussion should acknowledge the transformed nature of the analyses when making interpretations.**

Yes, thank you for catching this. We did indeed apply a parametric z-transformation to all variables, including the response variable. Scaling does not affect the model selection, but allows interpretation of the estimated slopes of the mixed effects models. We now give motivation for this in section 2.5.3 (Lines 244-247).

The rank transformation (Equation 1) derived normalized spatial deviation from the median is not done to satisfy model assumptions, but to facilitate comparison between events. It allows to decompose between effect of temporal (event median) and spatial variation (characterized by $\delta X$). Both are independent from each other and both are used in parallel as predictors in the statistical model. In doing so we can test whether, for example enhanced soil moisture response is due to either (i) the local throughfall hotspots or (ii) a stronger rainfall event. In case (i) we expect a significant effect of $\delta P_{TF}$ and in (ii) a significant effect of the spatial median $\hat{P}_{TF,j}$. As all variables entering the mixed effects model, all $\delta X$ (X being the variable, e.g. $P_{TF}$, $\theta_{pre}$, $\theta_{post}$) were z-transformed. We now explain this specifically in the text surrounding Eq. 1, where the delta values are introduced. (Lines 114-125) and where the soil water content variables are introduced (Lines 154-158).

**[R1.12] Tables 2, 3, and S5: except for the sign, it is redundant to present both t and p. It would be better to see effect size to help judge importance of these relationships, not just consistency of them. I do not understand what is literally meant by "variables are scaled such that the t-value indicates the effect strength" (Table 3 caption), because I do not know what "strength" means in this context. I think this sentence confuses consistency of the**

This is true and also relates to the main comment stating that the interpretation of the results is difficult. We therefore present results differently in the revision, showing the effect size (slope estimate) for each factor for the selected model. With all variables being z-transformed, the slope estimate gives an indicates the influence of the specific predictor on the response variable. To improve evaluation of the results, we also moved previous Tables 3 and 4 to the supplement (now Tables S5 and S6). We replaced them with in the main text with Fig. 5 giving the slope estimates. We hope that the visualization improves assessing the effect strength of the many different variables.

**[R1.13] Tables: what is the meaning of italics and bold?**

We have removed the italics. Bold mark significant effects. We also added the information to the figure and table captions.

**[R1.14] Fig 2 grouping by storm size is not necessary; please consider whether a scatterplot—with storm size as the independent variable—would give richer information.**

Yes, good point. We changed the figure (Figure 2) to scatter plot.

**[R1.15] L315 n=11 or n=3 as fig 4?**

Thank you for spotting this, now corrected on Line 296 and Line 302. The throughfall sample is larger, as we included also the events of 2014 when soil moisture sensors were not yet installed.

**[R1.16] L345 Table S6 is missing**

The table was removed, now we also erased the reference to the table.

**[R1.17] Fig 5-6 are the z scale transformations simply for these figures or are all statistical analyses also z scaled for the independent and dependent variables?**

Yes, all data were z-scaled. Now this is specifically mentioned in the Methods section (section 2.5.3, Lines 244-247) as well as in the Figure captions of Fig 5-7.

**[R1.18] L420 it would be good to show at least some data vs. kriged surfaces; readers have no way to know how to judge this now except this sentence in the discussion.**

We added a figure showing kriged surfaces of a small and a large event to the Supplement (Fig S4), to demonstrate how the number of sensor locations decreases when in large compared to small events.

**[R1.19] L422 the previous sentence says the real variation was not maintained.**

We meant to say that this can be an issue, but was apparently not a strong one with our data. We have reformulated (Lines 415 - 418)

**[R1.20] L426 the technique to estimate the variogram is irrelevant; the question is whether the assumptions of Kriging are useful in this application. Neither Voss nor Lark established that it is, so I do not agree with citing them here. Just because the data can be modeled by variograms does not mean Kriging assumptions hold, and there are good reasons to believe that throughfall does not vary smoothly in space. As the first attempt to use Kriging quantitatively for interpolation, the burden is on the current manuscript to establish the applicability of the technique. A more detailed discussion of the Kriging in light of what the literature says about spatial variability would be an**

**important addition. For example, one issue that is not addressed here is the role of sampler size in spatial variability estimates and interpolation.**

What we meant to say is that the number of samplers, the location design and the applied workflow adhere to the state of the art for derivation of the throughfall variogram that is next used for kriging. We reformulated this part. We now also mention the cross-correlation which yielded satisfactory results, and refer to sampler and sample size. Still, we tried keeping this section compact, partly to accommodate the other reviewer. (Lines 419-424)

**Technical corrections:**

Thank you for finding and typing those!

**L108 surely the small mountain does not move**

True, it is rather immobile. We replaced "of" with "on". Thank you.

**L124 minimize** - done

**L127 points** - done

**L127 the immediate vicinity** - done

**L164 depths** - done

**L166 location** - done

**L206 smaller than** - done

**L214 served -** done

**The Supplement table numberings are confusing: e.g., "Table S1 / S4."**

This was corrected, thank you!

**Fig S3 the axis is mislabeled and should be "CQV"**

Was corrected, thank you!

**L275 is a redundant Table caption and can be deleted. Same for L275, L283-284, L296-297, L338, L350, L359, L366.**

We agree that these short sentences stating the content of the tables are a repetition to the table caption. To our taste, this improves navigation through the text, and we prefer to keep as is.

**L332 "event sizes" -** done

**L333 "post-hoc test of a one-way ANOVA" isn't literally correct.**

We changed the wording (Lines 304-305).

**L359, L378 it would be clearer to say added, not stored** – done

**L379 I don't think "factors…were related to" is what was intended**

We changed the wording (now Line 386).

**L442 I don't understand "next to"**

Thank you! This was a false friend in translating German to English. We replaced it with "besides", "alongside", "as well as", "In addition to".

**L484 "weak"** - done

**Reviewer 2**

Thank you for the insightful feedback to the manuscript, and constructive recommendations. Below we respond to the main concerns to show how we are going to address them in the revision.

**[R2.1] The abstract is a little bit long, some of the results are duplicated as those regarding the effect of soil water distribution on soil water dynamic (lines 29-33 and lines 35-37so the authors can consider to shorten it..**

Done. We have shortened especially the results part of the abstract.

**[R2.2] There exist some confusion on using the terms δPTF ; δθpre and δθpost do they refer to normalized variables; spatial distribution , spatial pattern?**

Thank you for the comment. Both reviewers found that the normalization/transformation of the spatially distributed variables and the use of the term "pattern" were not well established. δ indicates the normalized deviation from the spatial median as shown in Equation 1, e.g.

$$\delta X_{ij} = \frac{X_{ij} - \hat{X}_j}{X_j}$$

where $X_{ij}$ is the spatially distributed variable, i the location, j the event, and $\hat{X}_j$ the jth event spatial median. This allows decomposing $X_{ij}$ into its magnitude (median, $\hat{X}_j$) and spatial deviation from the median ($\delta X_{ij}$). Both are independent from each other and both are used in parallel as predictors. We refer to $\delta X$ (with indices dropped) as the "spatial pattern", because "normalized deviation from the spatial median" seemed too long and bulky. We now motivate this specifically in the text surrounding Eq. 1, where the transformation/normalization is introduced. (Lines 119-133) and one more time when we introduce variables related to soil water content (Lines 154-159). We also added Table 1, that summarizes all variables, indices and operation symbols used in the statistical model. We also now mention that we drop indices for simplicity.

**[R2.3] Data on event topsoil seems in figure S3 seem not to follow a clear concave decreasing relationship with mean soil water content as said in line 298-299. Indeed, the variation around SWC equal to 20 is positive. I hesitate these results support the statements of lines 298-299. As it is no longer discussed in the paper, it might be deleted. What does it mean "enhance" soil water content variation?**

Agreed, in very dry conditions the spatial variation of post-event soil water content decreases again. The latter can only partly be attributed to the fact that rain events were small. However, we would like to keep this section, but reformulate it. Here we want to mainly illustrate that post event soil water content has a substantially higher spatial variation than the pre-event, especially in drier soils. Thus, while the absolute values of soil water content are little related to the throughfall input patterns, spatial variation increases substantially due to the rain events. This suggests that different factors interact in a way that cannot be completely revealed by the statistical model. We reformulated this section to refer to "moderately dry soils" instead of "dry soils". This high variation mentioned here is referred to again in the discussion now in Lines 470-474.

On the y-axis, the typo was corrected in the revision

**[R2.4] Discussion of GLM model on the soil water response is rather complex. Particularly interpretation of the interactions of fixed effects in lines 376-378 and figure 6. A clearer a more detailed explanation (also in the legend of figure 6) would make easier to the reader understand the foundations of the final statement. "Locally drier soil increased soil water storage in wet, but decreased it in dry times"**

Thank you for the feedback and advise. Agreed, we admittedly struggled when formulating this section. We reformulated this paragraph mentioned lines (now lines 370-373), hopefully improving the communication. We also added explanations to the figure captions of now Figures 6 and 7 as proposed by the reviewer. We also show the groups Fig 6 and 7 as the real spatial medians (back transformed median throughfall and volumetric soil water content). In Figure 7 this now shows specifically the different soil water conditions in topsoil and subsoil.

**[R2.5] Section 3.2 is entitled "spatial pattern of throughfall" while it also includes results on soil water content variation.**

Thanks. Done. Header of section 3.2 was changed to "Spatial pattern of throughfall and soil water content"

**Technical corrections**

**Line 53, line 518 and others: what do the authors want to say by using "next to" ?**

Thank you! This was a false friend in translating German to English. We replaced it with "besides", "alongside", "as well as", "In addition to".

**Line 484:  I think it is weak instead of week**

Done. Thanks.
* * *
References:

Demir, G., Guswa, A. J., Filipzik, J., Metzger, J. C., Römermann, C., and Hildebrandt, A.: Root water uptake patterns are controlled by tree species interactions and soil water variability, Hydrol. Earth Syst. Sci. Discuss. [preprint], https://doi.org/10.5194/hess-2023-91, in review, 2023.

Hupet, F., Lambot, S., Feddes, R. A., van Dam, J. C., and Vanclooster, M.: Estimation of root water uptake parameters by inverse modeling with soil water content data, Water Resources Research, 39, 2003.

Jackisch, C., Knoblauch, S., Blume, T., Zehe, E., and Hassler, S. K.: Estimates of tree root water uptake from soil moisture profile dynamics, Biogeosciences, 17, 5787–5808, https://doi.org/10.5194/bg-17-5787-2020, 2020.